# Feature Flow Regularization: Improving Structured Sparsity in Deep Neural Networks

## Abstract

Pruning is a model compression method that removes redundant parameters and accelerates the inference speed of deep neural networks (DNNs) while maintaining accuracy. Most available pruning methods impose various conditions on parameters or features directly. In this paper, we propose a simple and effective regularization strategy to improve the structured sparsity and structured pruning in DNNs from a new perspective of evolution of features. In particular, we consider the trajectories connecting features of adjacent hidden layers, namely feature flow. We propose feature flow regularization (FFR) to penalize the length and the total absolute curvature of the trajectories, which implicitly increases the structured sparsity of the parameters. The principle behind FFR is that short and straight trajectories will lead to an efficient network that avoids redundant parameters. Experiments on CIFAR-10 and ImageNet datasets show that FFR improves structured sparsity and achieves pruning results comparable to or even better than those state-of-the-art methods.

## 1 Introduction

Deep neural networks (DNNs) have achieved huge success in a wide range of applications. Meanwhile, DNNs require considerably more computational cost and storage space as they become deeper in order to achieve higher accuracy. Denil et al. (2013) demonstrated that there is significant redundancy in the parameterization of DNNs. The Lottery Ticket Hypothesis (Frankle & Carbin, 2019) conjectures that there exist sparse sub-networks that can obtain a comparable accuracy with the original network when trained in isolation.

Model compression methods have been proposed to balance accuracy and model complexity, e.g. weight pruning (Drucker & Le Cun, 1992; Hassibi & Stork, 1993; Han et al., 2015; Guo et al., 2016; Hu et al., 2016; Li et al., 2016) and quantization (Gong et al., 2014), low-rank approximation (Denton et al., 2014; Jaderberg et al., 2014; Liu et al., 2015), and sparsity structure learning (Wen et al., 2016; Alvarez & Salzmann, 2016; Zhou et al., 2016; Liu et al., 2017; Louizos et al., 2018; Gao et al., 2019; Yuan et al., 2020). Weight pruning removes less important parameters in the network. In particular, filter pruning (Li et al., 2016) removes entire filters in the network together with their related channels, which can compress and accelerate DNNs efficiently.

Existing structured pruning methods can be divided into two categories: parameter-based methods (Li et al., 2016; Molchanov et al., 2016; Liu et al., 2017; He et al., 2018; Lin et al., 2019; He et al., 2019; Zhuang et al., 2020; Liebenwein et al., 2020) that use some criteria to identify unimportant filters and remove them, and feature-based methods (Luo et al., 2017; He et al., 2017; Zhuang et al., 2018; Ye et al., 2018; Li et al., 2020; Lin et al., 2020; Tang et al., 2020) that select unimportant feature maps and then remove related filters and channels. For example, Li et al. (2020) incorporated two feature map selections: discovering features with low diversity and removing features that have high similarities with others.

In this paper, we propose a new regularization method on the trajectory connecting features of adjacent hidden layers, namely feature flow regularization (FFR). FFR smooths the trajectory of features, which implicitly improves the structured sparsity in DNN. Our motivation is that the trajectory of data along the network reflects the DNN structure. Shorter and straighter trajectory corresponds to an efficient and sparse structure of DNN. An illustration is given in Figure 1b.

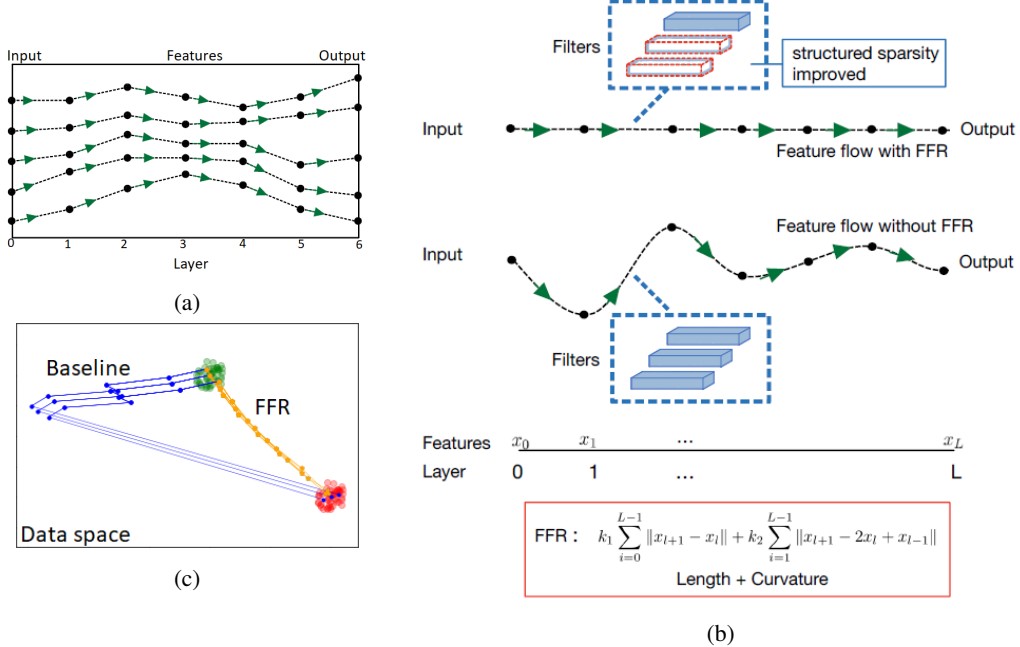

Figure 1: (a) Feature flow demonstration: Each curve represents a feature flow, i.e., the trajectory connecting the features of hidden layers, where nodes are input, features and output. (b) Feature flow regularization demonstration: Feature flow under FFR is shorter and straighter due to the length and curvature penalty. As a result, FFR improves the structured sparsity and leads to effective filter pruning. (c) An illustration example in two dimensional space showing the smooth effect of FFR, which is a five-block ResNet trained with FFR and without FFR (the baseline). The input data, features and targets are all points in two dimensions, and the feature flows are curves in two dimensions. The green cluster and red cluster contain input data points and the targets, respectively.

Our main contributions are: (1) We propose a new regularization (FFR) on the trajectory connecting the features of hidden layers, to improve the structured sparsity in DNN from a perspective of the trajectory of data along the network. This method is different from the existing sparsity structure learning methods, which directly impose regularization or constraints on the parameters. Our method is also different from those pruning methods based on feature maps, which use the information of the feature map individually or in pairs (for similarity) without global relationship. (2) We analyze the effect of FFR applied to convolutional layer and residual layer, and show that FFR encourages DNN to learn a sparse structure during training by penalizing the sparsity of both parameters and features. (3) Experimental results show that FFR achieves a comparable or even better pruning ratio in terms of parameters and FLOPs than recent state-of-the-art pruning methods.

## 2    RELATED WORK

**Filter pruning.**    Various criteria for filter selection in pruning have been proposed. Li et al. (2016) used $L_1$ norm to select unimportant filters and removed the filters whose norm is lower than the given threshold together with their connecting feature maps. Molchanov et al. (2016) measured the importance of filters based on the change in the cost function induced by pruning. Luo et al. (2017); He et al. (2017) formulated pruning as a constraint optimization problem and selected most representative neurons based on minimizing the reconstitution error. Lin et al. (2019) pruned filters as well as other structures by generative adversarial learning. He et al. (2019) pruned redundant filters utilizing geometric correlation among filters in the same layer. Hu et al. (2016); Zhuang et al. (2018); Li et al. (2020); Lin et al. (2020) removed filters based on the information, e.g. sparsity, rank or diversity, of feature maps that are generated by the filters. Our method learns sparse DNN during training and adopts magnitude-based pruning scheme after training.

**Sparsity regularization.** Some studies introduced sparsity regularization to find sparse structure of DNN. A commonly used strategy is to impose group Lasso and relaxed $L_0$ regularization (Zhou et al., 2016; Wen et al., 2016; Alvarez & Salzmann, 2016; Louizos et al., 2018). Liu et al. (2017); Huang & Wang (2018) associated a scaling factor with feature maps and imposed regularization on these scaling factors during training to automatically identify unimportant channels. The feature maps with small scaling factor values will be pruned. Gao et al. (2019) imposed a Cross-Layer grouping and a Variance Aware regularization on the parameters to improve the structured sparsity for residual models. Zhuang et al. (2020) used polarization regularizer on scaling factors. Yuan et al. (2020) proposed a method to dynamically grow deep networks by continuously sparsifying structured parameter sets. Different from these available methods that directly introduce regularization on the parameters or scaling factors, our FFR imposes regularization on the trajectory connecting features of hidden layers to control the parameters and enforce structured sparsity implicitly.

## 3 METHOD

### 3.1 FEATURE FLOW REGULARIZATION

We define **feature flow** as the trajectory formed by connecting the output features of adjacent hidden layers. For a DNN, the collection of trajectories with different input data from the training dataset reflect the network structure. We control these trajectories to obtain a sparse network.

Consider the forward propagation of a DNN with $L$ layers $\{x_l\}_{l=0,1,...,L}$:

$$x_{l+1} = h_l(x_l, w_l), \tag{1}$$

where $x_{l+1}$ is the output feature of the $l$-th layer, $h_l$ is the mapping in the $l$-th layer, and $w_l$ is the collection of trainable parameters. Introducing a temporal partition: $\{t_l = l/L\}_{l=0}^{L}$ with time interval $\Delta t = 1/L$, and regarding $x_l$ as the value of a function $x(t)$ at time step $t_l$, without considering dimensional consistency, Eq. (1) can be rewritten as (He et al., 2016; E, 2017; Lu et al., 2017; Chen et al., 2018)

$$x(t_{l+1}) = x(t_l) + \Delta t \cdot \hat{h}_l(x(t_l), w_l), \tag{2}$$

where $\hat{h}_l = (h_l - x_l)/\Delta t$. This can be interpreted as a discretization of evolution along a trajectory of the network described by an ordinary differential equation (E, 2017; Lu et al., 2017; Chen et al., 2018)

$$\frac{dx(t)}{dt} = \hat{h}(x(t), w(t), t). \tag{3}$$

The feature flow is the trajectory formed by connecting features of $\{x_l\}$, and is denoted by $\Gamma\{x_l\}$. See Figure 1a for an illustration of the feature flow.

We regard the trajectory $x(t)$ as well as the feature flow trajectory $\Gamma\{x_l\}$ as a "curve". Recall that for a curve $\gamma(t) : [0,1] \to R^D$ with arc length parameter $s$, its length $C(\gamma)$ and total absolute curvature $\mathcal{K}(\gamma)$ are

$$C(\gamma) := \int_0^1 \|\gamma'(t)\| dt, \quad \mathcal{K}(\gamma) := \int_0^{C(\gamma)} |\kappa(s)| ds, \tag{4}$$

where $\kappa(s) = \|\gamma''(s)\|$ is the curvature of the curve.

We introduce **feature flow regularization (FFR)** to improve the structured sparsity of DNN, borrowing the definitions of length and total absolute curvature of a curve to the feature flow, i.e., the trajectory formed by connecting features of hidden layers. For a feature flow associated with hidden layer features $\{x_l\}_{l=0,1,...,L}$, the FFR is

$$\mathcal{R}(x) := k_1 C(x) + k_2 \mathcal{K}(x), \tag{5}$$

where

$$C(x) = \sum_{l=0}^{L-1} \|x_{l+1} - x_l\|, \quad \mathcal{K}(x) = \sum_{l=1}^{L-1} \|x_{l+1} - 2x_l + x_{l-1}\|, \tag{6}$$

with $\|\cdot\|$ being the $L_1$ norm, and $k_1, k_2 > 0$ the hyperparameters. Here up to some constant factors, $C(x) \sim \sum_{l=0}^{L-1} \|x'(t_l)\|$ is an approximation of the total length of the trajectory $x(t)$, and

$\mathcal{K}(x) \sim \sum_{l=1}^{L-1} |\kappa(t_l)|$ is an approximation of its total absolute curvature, where $\{x_l = x(t_l)\}$ is a discretization of trajectory $x(t)$ with time partition $\{t_l = l/L\}_{l=0}^{L}$.

FFR smooths the feature flow by controlling the length and curvature. Intuitively, the length term in Eq. (6) makes the feature flow short, and the curvature term in Eq. (6) keeps the feature flow from bending too much. As a result, DNN trained under FFR has a more sparse structure; See demonstration in Figure 1b. More quantitative analysis is given in Sec. 4. We further give an illustration example in Figure 1c, which is a two-dimensional visualization showing the smooth effect of FFR on the feature flow of a ResNet. In this example, the input, features and output are all points in two dimensions, and the feature flows are actual curves in two dimensions. We can see that the feature flow under FFR is shorter and straighter. More detail of this illustration example is given in Appendix Sec. A.1. The curvature term also encourages more uniform distribution of hidden features on the trajectory from the input to the output, which makes it easier and more accurate to learn with sparse parameters; see the discussion in appendix A.4 and Figure 7

## 3.2 FFR APPLIED TO DNN AND PRUNING

For a DNN and training dataset $\left\{(x^{(j)}, y^{(j)})\right\}_{j=1}^{N}$, using Eq. (5), the FFR is:

$$\mathcal{R}(\mathbb{X}^{(j)}) = k_1 \sum_{i=0}^{L-1} \|x_{l+1}^{(j)} - x_l^{(j)}\| + k_2 \sum_{i=1}^{L-1} \|x_{l+1}^{(j)} - 2x_l^{(j)} + x_{l-1}^{(j)}\|, \tag{7}$$

where $\mathbb{X}^{(j)} = \{x_l, l = 0, \ldots, L\}^{(j)}$ denotes the set of output features of hidden layers for the $j$-th input data. The loss function with FFR in training is:

$$\frac{1}{N} \sum_{j=1}^{N} \left[ \mathcal{J}(x^{(j)}, y^{(j)}, W) + \mathcal{R}(\mathbb{X}^{(j)}) \right], \tag{8}$$

where $\mathcal{J}(x, y, W)$ is the loss function before applying FFR.

Note that two hidden states $x_l, x_{l+1}$ may have different dimensions. To fix the dimension mismatch problem, we adopt the same strategy as He et al. (2016), i.e, using a linear projection $\mathbf{P}_l$ by the shortcut connections to match the dimensions. We replace $x_l$ in Eq. (7) by $\mathbf{P}_l x_l$, where $\mathbf{P}_l$ is the learnt projection matrix and will be treated as learnable parameters in training. In implementation, we first group the features according to the stage (features dimensions):

$$\mathbb{X}^{(j)} = \bigcup_{g=1}^{G} \{x_{g,1}, x_{g,2}, \ldots, x_{g,l_g}\}^{(j)}, \quad L = \sum_{g=1}^{G} l_g. \tag{9}$$

Here $G$ is the number of stages in $\mathbb{X}^{(j)}$ and $l_g$ is the number of hidden layers in stage $g$. Secondly, we use the projection matrix to link different groups since the dimensional mismatch only occurs at the first feature of each stage. Using this method, the FFR of $\mathbb{X}^{(j)}$ becomes:

$$\mathcal{R}(\mathbb{X}^{(j)}) = k_1 \sum_{g=1}^{G} \left[ \|x_{g,1}^{(j)} - \mathbf{P}_g x_{g-1,l_{(g-1)}}^{(j)}\| + \sum_{i=1}^{l_g-1} \|x_{g,i+1}^{(j)} - x_{g,i}^{(j)}\| \right]$$

$$+ k_2 \sum_{g=1}^{G} \left[ \|x_{g,2}^{(j)} - 2x_{g,1}^{(j)} + \mathbf{P}_g x_{g-1,l_{(g-1)}}^{(j)}\| + \sum_{i=2}^{l_g-1} \|x_{g,i+1}^{(j)} - 2x_{g,i}^{(j)} + x_{g,i-1}^{(j)}\| \right]. \tag{10}$$

In the meantime, the hyperparameters $k_1, k_2$ may vary with the feature dimensions so that FFR can uniformly control the features at different stages. In our experiments, we adjust $k_1, k_2$ to be inversely proportional to the scale of feature maps. Moreover, the FFR process can be generalized to the case where we choose features every several layers. We can denote the selected hidden layers as $\mathbb{L} = \{l_i, i = 0, 1, \ldots, m, m = \#\mathbb{L}\}$. Then we apply FFR to the feature flow that connects the features in the set $\mathbb{X}^{(j)} = \{x_l, l \in \mathbb{L}\}$.

After training, we conduct one-shot filter pruning: removing filters with small magnitude and removing channels in the next layer that convolve with the feature maps generated by the pruned filters. Finally, we fine tune the pruned network for a few epochs.

Our FFR training and pruning method is summarized in Algorithm 1.

---

**Algorithm 1** FFR Training and One-shot Pruning

---

**Require:** training dataset $\left\{(x^{(j)}, y^{(j)})\right\}_{j=1}^{N}$, a neural network and hyperparameters $k_1, k_2$.
  **Pre-step 1:** group the features in $\mathbb{X} = \{x_l, l \in \mathbb{L}\}$ according to the stage as in Eq. (9),
  **Pre-step 2:** write down FFR $\mathcal{R}(\mathbb{X}^{(j)})$ in Eq. (10) for each paired data $(x^{(j)}, y^{(j)})$.
  **Training step:** train the network under loss function with FFR given in Eq. (8).
  **Pruning step:** remove filters and the corresponding channels in the trained model,
  fine tune the pruned model for a few epochs.
  **return** a compact neural network.

---

### 3.3 FEATURE FLOW OF VGGNET AND RESNET

In this subsection, we demonstrate how to construct feature flows of VGGNet (Simonyan & Zisserman, 2015) and ResNet (He et al., 2016), which are two commonly used network architectures. Similar construction can apply to other neural networks.

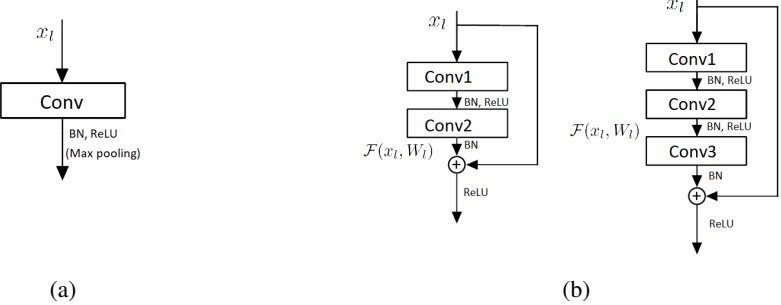

(a)                                                                    (b)

Figure 2: (a) A convolutional block in VGGNet. (b) Residual blocks in ResNet.

**VGGNet.** VGGNet is a convolutional neural network with a plain structure. Each convolutional layer is followed by some activation functions: batch normalization, rectified linear activation function (ReLU), and sometimes a maxpooling layer. We regard the convolutional layer with all its activations as a block; see Figure 2a. We collect the output of every convolutional block in VGGNet to form the feature flow $\Gamma\{x_l\}$. Let $x_l$ be the output feature of the $l$-th block, then the feature of the $l + 1$-th convolutional block is

$$x_{l+1} = (\text{Max pooling}) \cdot \sigma(\mathcal{BN} \odot (W_l \otimes x_l)), \tag{11}$$

where $W_l$ is the convolutional weights, $\mathcal{BN}$ is batch normalization, and $\sigma(\cdot)$ is ReLU. Max pooling only appears once every several layers. $\otimes$ denotes convolution operation and $\odot$ denotes batch normalization operation.

**ResNet.** ResNet takes residual blocks as building blocks, each of which contains two or three convolutional layers; see Figure 2b. We collect the output of every residual block to form the feature flow $\Gamma\{x_l\}$. The feature of $l + 1$-th residual block is

$$x_{l+1} = \sigma(\mathcal{F}(x_l, W_l) + W_{s_l} x_l), \tag{12}$$

where $\mathcal{F}(x_l, W_l)$ is the residual function, $W_l = \{W_{l,k} | k = 1, \dots, K\}$ is the set of convolutional weights, $K = 2$ or $3$ is the number of convolutional layers in each residual unit. $W_{s_l}$ is the identity matrix if $x_l$ and $\mathcal{F}(x_l, W_l)$ are in the same dimension, and the learnt projection matrix otherwise.

## 4 SPARSITY ANALYSIS

In this section, we demonstrate how FFR improves structured sparsity. As explained in the previous section, the idea of FFR is to shorten and straighten the trajectory of the input along the network. We will show that such effect of FFR encourages feature sparsity in addition to penalizing the sparsity of

the parameters. Sparse features contribute to structured sparsity: for zero-value feature map, we can remove the filter which produces the feature map and the channel which convolves with the feature map. Therefore, FFR improves the structured sparsity in the network.

### 4.1 FEATURE AND PARAMETER SPARSITY

We analyze the effect of FFR when it is applied to the output features of the convolutional block and the residual block, which are commonly used in DNNs. We focus on the length term in Eq. (6) in FFR.

**Convolutional block.** Ignoring the activation functions in Eq. (11), the length term in FFR is reduced to

$$\|x_{l+1} - x_l\| = \|W_l \otimes x_l - x_l\| = \|(W_l - I) \otimes x_l\|, \tag{13}$$

where $I$ denotes the 'identity' convolution kernels: for the $i$-th filter, only the $i$-th channel is a nonzero matrix and all other channels are zero-matrices. This length term pushes the parameter $W_l$ to be close to the identity kernel $I$, which is highly structured sparse. In the meantime, the length term also pushes the feature $x_l$ to be sparse. Figure 3a shows the $L_1$ norm of 512 feature maps generated by the last convolutional layer in VGG16 trained with and without FFR on CIFAR-10. As shown in the figure, the baseline network (VGG16 trained without FFR) has few zero-valued feature maps. In contrast, the network trained under FFR learns much more sparse features: most feature maps have zero-valued norm that can be removed. Such significant improvement of feature sparsity under FFR is also observed in the comparison for other layers of VGG16; see Appendix section A.2 for more plots.

**Residual block.** The length term in FFR enhances sparsity of the residual block by pushing the residual function $\mathcal{F}(x_l, W_l)$ and features $x_l$ to zeros. We ignore the projection matrix $W_s$ in the shortcut connection since it only appears when the stage changes. From Eq. (12), the length term in FFR is reduced to

$$\|x_{l+1} - x_l\| = \|\sigma(\mathcal{F}(x_l, W_l) + x_l) - x_l\| \stackrel{\text{element-wise}}{=} \begin{cases} \|\mathcal{F}(x_l, W_l)\|, & \mathcal{F}(x_l, W_l) + x_l \geq 0, \\ \|x_l\|, & \mathcal{F}(x_l, W_l) + x_l < 0. \end{cases} \tag{14}$$

Under the hypothesis that the optimal function is closer to an identity mapping than to a zero mapping (He et al., 2016), the first case $\mathcal{F}(x_l, W_l) + x_l \geq 0$ holds most of the time, and in this case, FFR penalizes the norm of the residual function. For a residual block with two or three convolutional layers, and ignoring the activations, the residual function can be reduced to

$$\mathcal{F}(x_l, W_l) = \sigma(\mathcal{BN}_2 \odot W_{l,2} \otimes \sigma(\mathcal{BN}_1 \odot W_{l,1} \otimes x_l)) \sim W_{l,2} \otimes W_{l,1} \otimes x_l, \text{ or} \tag{15}$$

$$\mathcal{F}(x_l, W_l) = \sigma(\mathcal{BN}_3 \odot W_{l,3} \otimes \sigma(\mathcal{BN}_2 \odot W_{l,2} \otimes \sigma(\mathcal{BN}_1 \odot W_{l,1} \otimes x_l))) \sim W_{l,3} \otimes W_{l,2} \otimes W_{l,1} \otimes x_l. \tag{16}$$

Hence imposing penalty on the residual function can make both parameters and features sparse. In the second case, where $\mathcal{F}(x_l, W_l) < -x_l$, FFR encourages the feature $x_l$ to be sparse.

Further details of how FFR improves the structured sparsity are given in appendix A.3.

Figure 3b shows the $L_1$ norm of 64 feature maps of the first residual block in ResNet56 trained with and without FFR on CIFAR-10. As shown in the figure, the network trained under FFR outputs more zero value feature maps than the baseline (trained without FFR). In Figure 3c, we visualize the parameters of shape $16 \times 16 \times 3 \times 3$ from the first convolutional layer in the first residual block of ResNet56 trained with FFR on CIFAR-10. We display the filter in row according to the magnitude: from top to bottom, the norm of the filter is increasing. It can be seen from the figure that the parameters of the network trained with FFR are of high structured sparsity. In the figure, small value filters are circled with a red horizontal square, and small value channels convolved with the same feature map are respectively circled with orange vertical squares.

In short, Figures 3a and 3b empirically prove that FFR improves structured sparsity of features, and Figure 3c shows its effect on the structured sparsity of the parameters.

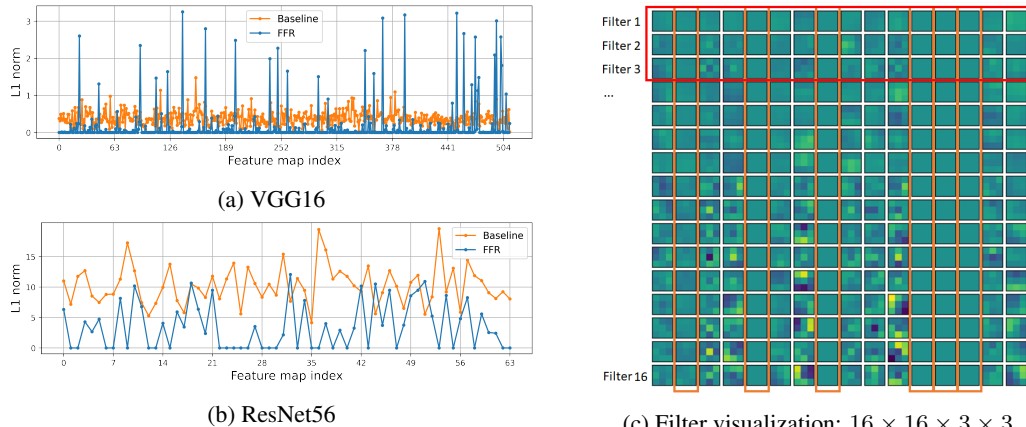

(a) VGG16

(b) ResNet56

(c) Filter visualization: $16 \times 16 \times 3 \times 3$.

Figure 3: (a) VGG16 feature maps trained with and without FFR: $L_1$ norm plot, with 512 feature maps in the feature of the last convolutional layer. (b) ResNet56 feature maps trained with and without FFR: $L_1$ norm plot, with 64 feature maps in the feature of the nineteenth residual bock. (c) Visualization of the filters in the first residual block in ResNet56 trained with FFR. From top to bottom, the filters are displayed according to their norm from small to large. Square denotes the filters (red, horizontal) and channels (orange, vertical) that have small magnitude and can be removed. Both VGG16 and ResNet56 are trained on CIFAR-10.

## 4.2 STRUCTURED SPARSITY IMPROVEMENT

To further show the ability of FFR in improving structured sparsity, we examine the relation between accuracy and structured sparsity. We compare VGG16 and ResNet56 with and without FFR trained on CIFAR-10. After training, we use an increasing threshold to zero the filters whose magnitude is under the threshold and the corresponding channels. Then we calculate the accuracy and structured sparsity that is defined as the percent of the parameters zeroed. Figure 4a and 4b show the accuracy-sparsity trade-off curves of VGG16 and ResNet56 on CIFAR-10, respectively. It can be seen that the network trained with FFR will not suffer accuracy degradation until it has a significantly large structured sparsity compared with the baseline case.

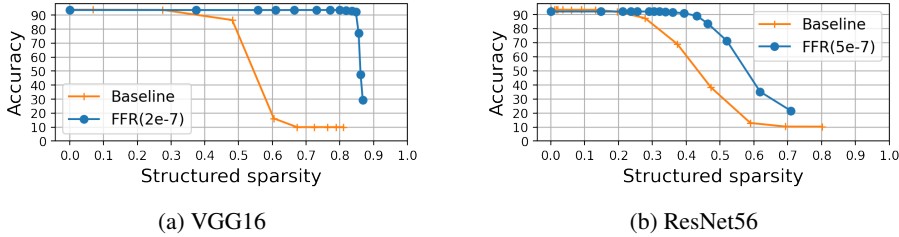

(a) VGG16

(b) ResNet56

Figure 4: Accuracy-sparsity trade off curve of (a) VGG16 and (b) ResNet56 trained with FFR (blue) and without FFR (orange, Baseline) on CIFAR-10.

## 5 EXPERIMENTS

### 5.1 IMPLEMENTATION

**Models and datasets.** To demonstrate the effectiveness of FFR in pruning, we perform experiments with VGGNet and ResNet on the CIFAR-10 (Krizhevsky & Hinton, 2009) and ImageNet (Deng et al., 2009) datasets. The VGG16 is a modified VGGNet in Simonyan & Zisserman (2015): we remove two fully-connected layers and only use one fully-connected layer for classification. We compare the network trained with FFR and without FFR as a baseline. We adopt weights initialization as in He et al. (2015).

**Training setting.**  FFR and baseline have the same training setting. For CIFAR-10, the mini-batch size is 128. We train VGG16 and ResNet56 200 epochs and set initial learning rate as 0.1 and divide it by 10 at the 80th, 120th, and 160th epoch. For ImageNet, the mini-batch size is 256. We train ResNet50 90 epochs and divide initial learning rate 0.1 by 10 at the 30th and 60th epoch. We use SGD optimizer with weight decay $5e^{-4}$ for CIFAR-10, $1e^{-4}$ for ImageNet, and momentum 0.9. Note that $L_2$ regularization is implemented by applying weight decay in SGD. We ran our experiments with pytorch.

**Pruning setting.**  We perform one-shot structured pruning: remove filters based on the magnitude and their corresponding channels in the next layer. For all experiments on CIFAR-10 and ImageNet, we only fine tune the pruned model 30 epochs with learning rate $1e^{-4}$ on CIFAR-10 and $1e^{-3}$ on ImageNet. For VGGNet, we prune the network directly, and for ResNet, we use zeros padding in the pruned dimension. The outputs of the shortcut and the last convolutional layer in the residual block in ResNet must have the same dimension because of the addition operation between them. Using zero padding, we can remove the filters flexibly. Moreover, if all filters of the first convolutional layer are pruned, then the residual block can be pruned.

### 5.2    STRUCTURED PRUNING

We compare pruning results of FFR with recent state-of-the-art pruning methods in Figure 5 and Tables 1 and 2. We present ablation study on the hyperparameters $k_1, k_2$ and a rule on hyperparameter selection in Appendix A.5.

**Results on CIFAR-10.**  Table 1 shows the pruning results comparisons of VGG16 and ResNet56 on CIFAR-10. For better comparison, in Figure 5, we plot error-parameter reduction and error-FLOPs reduction trade-off curves of VGG16 trained with FFR together with the results of other methods shown in Table 1. We can see that FFR is able to achieve larger pruning ratio in terms of parameters and FLOPs. Specifically, we prune VGG16 with 90.2% parameters reduction and 61.2% FLOPs reduction with a slight drop (0.66%) in accuracy, and ResNet56 at a large FLOPs reduction 60.6% with accuracy drop 1.05%. From the comparisons of VGG16 results in Figure 5, the FFR method has lower error reductions in parameter and FLOPs than most of the available methods, and achieves larger pruning ratio. Note that the DCP method (Zhuang et al., 2018) conducts channel pruning and fine tunes the network stage by stage to achieve lower error, whereas in FFR, we perform only one-shot structured pruning with a few epochs of fine tune. We also visualize the feature maps of the reserved and pruned filters of the first convolution layer in VGG16 using FFR in Appendix A.7, Figure 10.

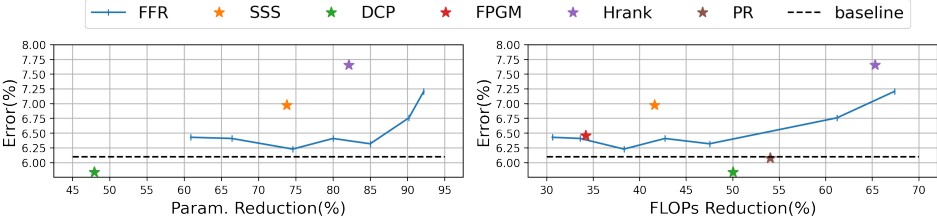

Figure 5: Error-parameter reduction and error-FLOPs reduction trade-off curves of VGG16 trained with FFR on CIFAR-10, and comparison with the results of SSS (Huang & Wang, 2018), DCP (Zhuang et al., 2018), FPGM (He et al., 2019), Hrank (Lin et al., 2020), PR (Zhuang et al., 2020).

**Results on ImageNet.**  Table 2 shows the pruning results comparisons of ResNet50 on ImageNet, which is a large-sacle dataset. Although there is only one-shot pruning and a few epochs fine-tuning in FFR, our FFR pruned ResNet50 achieves comparable parameters and FLOPs reductions with the recently proposed pruning methods.

**Methodology comparison.**  Methodologically, our FFR method manipulates the trajectory connecting features of hidden layers, which is different from pruning methods based on feature maps

information, e.g. Hu et al. (2016); Li et al. (2020); Lin et al. (2020) and is different from methods that impose regularization the parameters, e.g. Huang & Wang (2018); Zhuang et al. (2020). In terms of implementation simplicity, we use SGD during training and adopt a one-shot pruning strategy instead of additional optimization steps (e.g., He et al. (2019)) or iterative pruning (Zhuang et al., 2018).

Table 1: Pruning results on CIFAR-10. '-' represents result not reported. 'M' represents $1e^6$.

| Model | Method | Baseline Error (%) | Pruned Error (%) | Param. Reduction | FLOPs Reduction |
|---|---|---|---|---|---|
| VGG16 | Baseline (Ours) | 6.10 | 6.10 | 0% (14.72M) | 0% (313M) |
| | SSS (Huang & Wang, 2018) | 6.10 | 6.98 | 73.8% | 41.6% |
| | DCP (Zhuang et al., 2018) | 6.01 | 5.84 | 47.9% | 50.0% |
| | FPGM (He et al., 2019) | 6.42 | 6.46 | - | 34.2% |
| | Hrank (Lin et al., 2020) | 6.04 | 7.66 | 82.1% | 65.3% |
| | PR (Zhuang et al., 2020) | 6.12 | 6.08 | - | 54.0% |
| | FFR ($k_1, k_2 = 2e^{-7}$) | 6.10 | 6.76 | 90.2% | 61.2% |
| ResNet56 | Baseline (Ours) | 6.60 | 6.60 | 0% (0.86M) | 0% (126M) |
| | CP (He et al., 2017) | 7.20 | 8.20 | - | 50.0% |
| | DCP (Zhuang et al., 2018) | 6.20 | 6.51 | 49.2% | 49.7% |
| | SFP (He et al., 2018) | 6.41 | 6.65 | - | 52.6% |
| | FPGM (He et al., 2019) | 6.41 | 6.51 | - | 52.6% |
| | Hrank (Lin et al., 2020) | 6.74 | 6.83 | 42.4% | 50.0% |
| | PR (Zhuang et al., 2020) | 6.20 | 6.17 | - | 47.0% |
| | FFR ($k_1, k_2 = 1e^{-7}$) | 6.60 | 7.65 | 49.7% | 60.6% |

Table 2: Pruning results on ImageNet. '-'represents result not reported. 'M' represents $1e^6$.

| Model | Method | Top-1 Err. Increase (%) | Top-5 Err. Increase (%) | Param. Reduction | FLOPs Reduction |
|---|---|---|---|---|---|
| ResNet50 | Baseline (Ours) | 0 (28.44) | 0 (9.72) | 0% (25.56M) | 0% (4089M) |
| | SSS (Huang & Wang, 2018) | 4.30 | 2.07 | 38.8% | 43.0% |
| | DCP (Zhuang et al., 2018) | 1.06 | 0.61 | 51.5% | 55.6% |
| | SFP (He et al., 2018) | 14.01 | 8.27 | - | 41.8% |
| | FPGM (He et al., 2019) | 1.32 | 0.55 | - | 53.5% |
| | Hrank (Lin et al., 2020) | 4.17 | 1.86 | 62.1% | 46.0% |
| | PR (Zhuang et al., 2020) | 0.52 | - | - | 54.0% |
| | FFR ($k_1, k_2 = 5e^{-8}$) | 2.68 | 1.40 | 52.3% | 47.2% |

## 6 CONCLUSIONS AND DISCUSSION

In this paper, we propose a simple and effective regularization method (FFR) from a new perspective of the data trajectory along the network. FFR smoothes the trajectory by imposing controls on the length and total absolute curvature of the feature flow, leading to significant increase of structured sparsity in DNNs. We perform a sparsity analysis of FFR for VGGNet and ResNet to validate this method. Experimental results show that FFR can significantly enhance structured sparsity, which enables us to prune filters efficiently in one pass. Future work may include: (1) perform rigorous sparsity analysis for FFR, and (2) integrate FFR with other pruning methods to further improve pruning results. The FFR method is a one-shot pruning method, in which the regularization is imposed on features instead of predetermining the sparsity. (See more discussion in appendix A.9.) Ablation study has been performed and a rule has been summarized on selecting the hyperparameters. The framework of feature flow proposed in this paper can be extended to other applications beyond pruning, e.g., network generalization (Drucker & Le Cun, 1992; Raghu et al., 2017).

REPRODUCIBILITY STATEMENT

The details of experiments are given in section 5.1.

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

## A    APPENDIX

### A.1    SETTING OF THE ILLUSTRATION EXAMPLE IN SECTION 3.1

To visualize the smoothing effect of FFR, we use a two-dimensional example. In Figure 1c, the green and red clusters contain 50 paired data points in two dimensional space. The green cluster is evenly distributed in a circle with center $(2, 6)$ and radius $0.5$. The red cluster is obtained by exactly shifting the green cluster four units to the right and four units down, that is in a circle with center $(6, 2)$. we use a ResNet to learn a translation mapping from the green cluster to the red cluster. The ResNet has five fully-connected residual blocks. Here each residual block contains two linear layers and outputs a two-dimensional feature. We train ResNet without and with FFR to learn the mapping under the same setting for comparison. A trajectory is obtained by starting from an input data point, connecting five output features and ending with the output. From Figure 1c, the trajectories of the network trained with FFR, represented by the orange curves, are shorter and more straight than those of the network trained without FFR, represented by the blue curves. We plot the feature flows of three test data for comparison. It shows that FFR indeed effectively smooths the feature flow.

### A.2    FEATURE MAPS NORM PLOT IN SECTION 4.1

We show the $L_1$ norm plot of 256 feature maps generated by the fifth convolutional layer and 512 feature maps generated by the eighth convolutional layer in VGG16 trained with and without FFR on CIFAR-10 in Figure 6a and 6b respectively. Consistent with the claim in the main text, the network trained under FFR learns much more sparse features with zero-valued norm than the baseline network (VGG16 trained without FFR). FFR improves the feature sparsity which contributes to the structures sparsity in DNN.

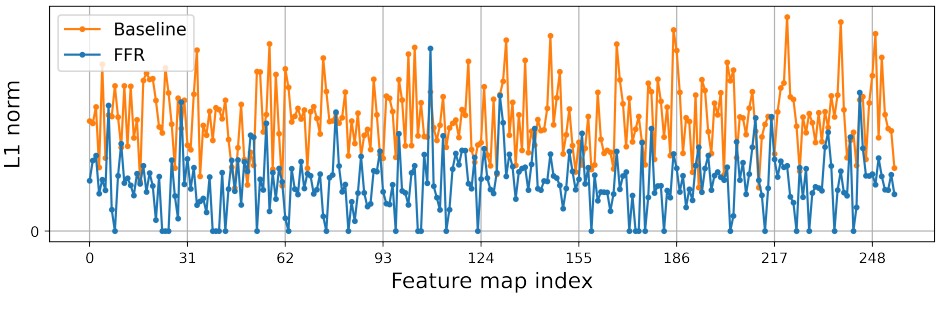

(a) 256 feature maps of the fifth convolutional layer.

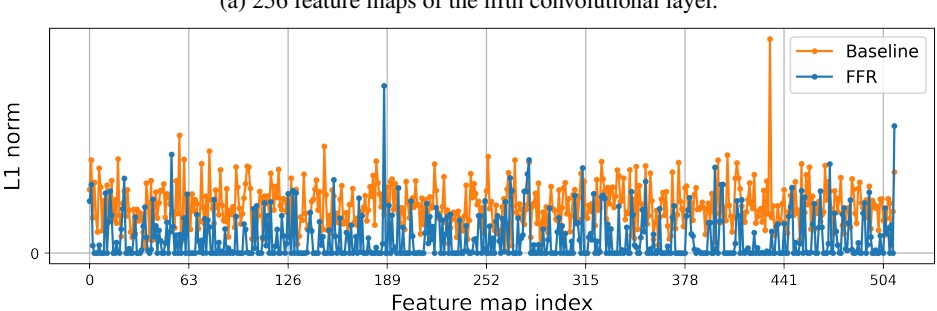

(b) 512 feature maps of the eighth convolutional layer.

Figure 6: VGG16 feature maps trained with and without FFR on CIFAR-10: $L_1$ norm plot.

### A.3 FURTHER SPARSITY ANALYSIS ON THE LENGTH TERM IN FFR

In Sec.4, we show that the proposed FFR method encourages structured sparsity of features in addition to penalizing the structured sparsity of the parameters. Here we give more details of how FFR improves the structured sparsity based on Eqs. (13)-(16) in Sec.4.

For $W \in R^{n \times c \times \omega \times h}, X \in R^{c \times a \times b}$, assume that $W \otimes X \in R^{n \times p \times q}$, where $\otimes$ denotes the convolutional operation, and $n, p, q$ are the dimensions of the output. We have

$$\|W \otimes X\| = \sum_{n,p,q} |W_{n,:,:,:} \otimes X_{:,a_p,b_q}| = \sum_{n,p,q} |\sum_c W_{n,c,:,:} \otimes X_{c,a_p,b_q}|, \tag{17}$$

where $\| \cdot \|$ denotes elementwise $L_1$-norm and $| \cdot |$ denotes the absolute value, and $X_{:,a_p,b_q} \in R^{c \times \omega \times h}$ represents the subset in features $X$ that produces the $(p, q)$-th entry $(W \otimes X)_{:,p,q}$ in each cannel of the output.

We first show that controlling $\|W \otimes X\|$ encourages both $W$ and $X$ to be small. Expanding tensor $W$ into matrix and tensor $X$ into vector, we rewrite the convolutional operation as matrix-vector multiplication:

$$\|W \otimes X\| = \|\hat{W}\hat{X}\|, \tag{18}$$

where $\| \cdot \|$ is the elementwise $L_1$-norm. Thus we have

$$\nabla_{\hat{X}} \|\hat{W}\hat{X}\| = \hat{W}^T \mathrm{sign}(\hat{W}\hat{X}), \tag{19}$$

where $\mathrm{sign}(\cdot)$ is the sign function elementwise applied to the matrix. It can be calculated that

$$\hat{X}^T(-\nabla_{\hat{X}} \|\hat{W}\hat{X}\|) = -\hat{X}^T \hat{W}^T \mathrm{sign}(\hat{W}\hat{X}) = -\|\hat{W}\hat{X}\| \leq 0. \tag{20}$$

Noticing that the direction of $\hat{X}$ is the increasing direction of the function $\frac{1}{2}\|\hat{X}\|^2 = \frac{1}{2}\|X\|^2$, or equivalently, the increasing direction of the norm $\|\hat{X}\| = \|X\|$, the above equation implies that minimizing $\|W \otimes X\| = \|\hat{W}\hat{X}\|$ leads to decrease in $\|X\|$. Similarly, we can have the conclusion that minimizing $\|W \otimes X\|$ leads to decrease in $\|W\|$.

In the meantime, under the regularization effects, in order to maintain the performance of the neural network as much as possible, the feature maps in $X$ tend to have only a few large norm feature maps (which capture the most representative information) and many zero norm feature maps, rather than feature maps with norms distributed evenly. One extreme example without expressiveness is noise which is uniform distribution. Therefore, the tradeoff between performance and sparsity of the network leads to high channelwise sparsity in features.

As a result, the convolutional parameters $W$ are trained to have high filterwise sparsity, since the filter $W_{n,:,:,:}$ that generates the zero feature map will become zero in training.

That is why controlling $\|W \otimes X\|$ encourages $W_{n,:,:,:}$ and $X_{c,:,:}$ to be small, which correspond to the parameters $W$ filter-wise sparsity and the feature $X$ channel-wise sparsity, respectively.

Using the claim above, the length term in FFR penalizes $\|(W_l - I) \otimes x_l\|$ for convolutional block and encourages high channel-wise sparsity of $x_l$ and high filter-wise sparsity of $W_l - I$, which leads to high structured sparsity of $W_l$. For the residual block, the length term in FFR penalizes $\|W_{l,2} \otimes W_{l,2} \otimes x_l\|$ or $\|x_l\|$, so it improves the structured sparsity of the convolutional parameters and the features. See Eqs.(13)-(16).

### A.4 DISCUSSION ON THE SPARSITY EFFECT OF THE CURVATURE TERM IN FFR

In fact, in addition to the effect that the curvature term keeps the trajectory from bending too much as shown in Sec.3.1 and Fig.1, the curvature term also encourages more uniform partition of the trajectory from the input to the output, which makes it easier and more accurate to learn with sparse parameters. This effect is associated with the "discretized" trajectory. This can be seen from the following illustrative example.

Consider two input-feature-output trajectories as in Fig. 7, where dots represent the input, the hidden features or the output, and the dashed lines denote segments on the trajectory whose length is

proportional to the $L_1$ distance. Here trajectories (a) and (b) have the same total length, so the length term cannot distinguish them. Whereas trajectory (a) has smaller "curvature term" than that of (b), thus it is preferred by the curvature term. In general, the longer $L_1$ distance between features of adjacent layers, the less possible it is for that layer to learn with sparse parameters.

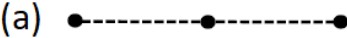

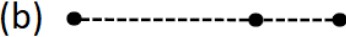

Figure 7: Illustration of the sparsity effect of the curvature term in FFR. Trajectory (a) is preferred by the curvature term over that in (b). The length is proportional to the $L_1$ distance between features in different layers.

## A.5 ABLATION STUDY ON HYPERPARAMETERS $k_1, k_2$

We tune the hyperparameters $k_1, k_2$ with experiments. The choice of hyperparameters depends on the architecture of the DNN and the scale of the input: for the DNN with more layers and higher resolution input. we use smaller $k_1, k_2$.

We perform ablation study on hyperparameters with different values. In Table 3, we report results of ablation study based on the pruning experiments of VGG16 on CIFAR-10. We use the same threshold to prune the network trained under FFR with different values of hyperparameters, and fine tune 30 epochs. We report error, parameters reduction and FLOPs reduction. We have also plotted accuracy-sparsity trade-off curves in Figure 8 for clearer comparison.

As shown in the table, the baseline (VGG16 trained without FFR) cannot preserve the accuracy under the same pruning and fine-tuning setting. From the table and Figures 8a and 8b, the curvature term has a stronger impact on the sparsity improvement, and in the mean time, higher error increase, than the length term. Compared with applying the length term or the curvature term alone, combining the length term and the curvature term can achieve higher accuracy and sparsity at the same time (see Figure 8c). Moreover, network trained under FFR with appropriate $k_1, k_2$ achieves large pruning ratio while maintaining the accuracy.

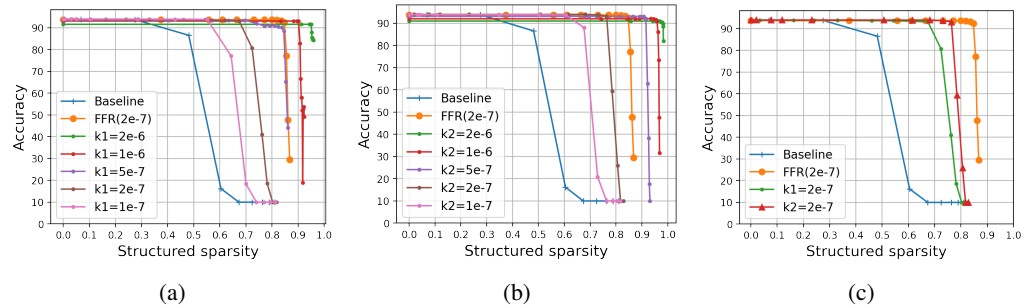

Figure 8: Accuracy-sparsity trade off curves of VGG16 trained under FFR with different hyperparameters on CIFAR-10: (a) $k_2 = 0$, different $k_1$, (b) $k_1 = 0$, different $k_2$, (c) $k_1, k_2$ work separately or together.

Table 3: Alation study on $k_1, k_2$. Pruning results of VGG16 on CIFAR10. 'M' represents $1e^6$.

| Hyperparamerer | Error(%) | Param. Reduction | FLOPs Reduction |
|---|---|---|---|
| Baseline | 6.10 | 0%(14.72M) | 0%(313M) |
| $k_1, k_2 = 0$ | 34.57 | 74.1% | 35.0% |
| $k_1 = 1e-7, k_2 = 0$ | 6.21 | 75.4% | 34.4% |
| $k_1 = 2e-7, k_2 = 0$ | 6.18 | 77.0% | 37.2% |
| $k_1 = 5e-7, k_2 = 0$ | 6.41 | 84.0% | 44.3% |
| $k_1 = 1e-6, k_2 = 0$ | 7.08 | 91.8% | 62.6% |
| $k_1 = 2e-6, k_2 = 0$ | 8.59 | 95.7% | 70.6% |
| $k_1 = 0, k_2 = 1e-7$ | 6.21 | 77.1% | 37.9% |
| $k_1 = 0, k_2 = 2e-7$ | 6.06 | 79.2% | 38.9% |
| $k_1 = 0, k_2 = 5e-7$ | 6.98 | 92.4% | 67.2% |
| $k_1 = 0, k_2 = 1e-6$ | 8.01 | 96.3% | 80.0% |
| $k_1 = 0, k_2 = 2e-6$ | 9.31 | 98.1% | 90.0% |
| $k_1, k_2 = 5e^{-8}$ | 6.36 | 76.5% | 36.4% |
| $k_1, k_2 = 1e^{-7}$ | 6.45 | 77.5% | 38.6% |
| $k_1, k_2 = 2e^{-7}$ | 6.32 | 85.0% | 47.5% |

A.6 MORE EXPERIMENTS ON STRUCTURED SPARSITY IMPROVEMENT

We further verify the effect of FFR method in terms of structured sparsity improvement using ResNet18, ResNet34 and ResNet50 on TinyImageNet. In Fig. 9, we plotted the accuracy-sparsity trade-off curves of FFR compared with the baseline similar to those in Fig. 4, and these figures also show that FFR improves the structured sparsity and maintains the accuracy in the experiments.

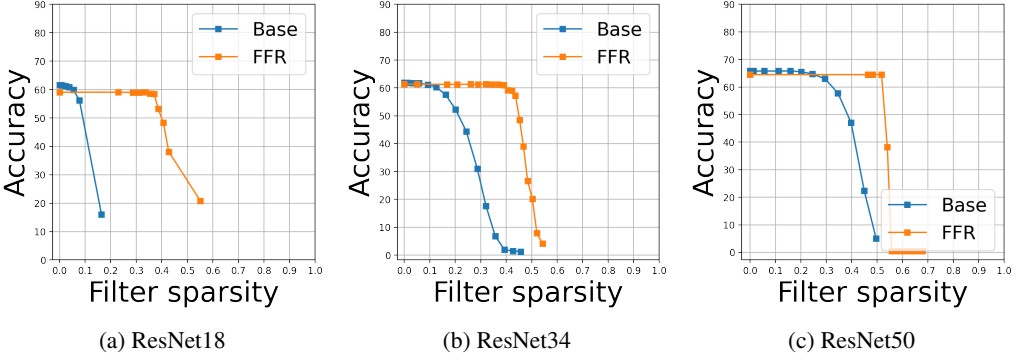

| (a) ResNet18 | (b) ResNet34 | (c) ResNet50 |
|---|---|---|

Figure 9: Accuracy-sparsity trade off curves of (a) ResNet18 (b) ResNet34 (c) ResNet50 trained with FFR and without FFR (Base) on TinyImageNet.

A.7 FEATURE MAPS VISUALIZATION

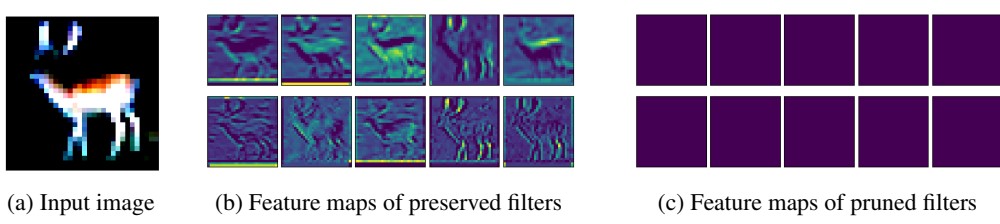

| (a) Input image | (b) Feature maps of preserved filters | (c) Feature maps of pruned filters |
|---|---|---|

Figure 10: Feature maps visualization of the first convolutional layers in VGG16 using FFR.

### A.8 FEATURE MAPS VISUALIZATION OF DIFFERENT LAYERS

We visualize the feature maps of different hidden layers in VGG16 with and without FFR on CIFAR-10. These figures show that in the baseline, the features of each layer contain various less informative feature maps that can be pruned, and FFR selects the most representative feature maps and lets other feature maps be zeros.

Therefore, FFR is indeed able to learn highly sparse structure while maintaining the representation power of extracting high-level features from low-level features.

### A.9 COMPARISONS WITH METHODS WITH PREDETERMINED PRUNING RATIO

Our FFR method performs one-shot pruning after training with regularization, and different pruning or sparsity ratios can be achieved by adjusting the hyperparameters instead of direct selecting the pruning ratio. Such indirect control of pruning ratio is common in many other one-shot regularization-based pruning methods such as SSS (Huang & Wang, 2018) and PR (Zhuang et al., 2020). In addition, we can also obtain the desired sparsity using different thresholds to prune the parameters in different layers in our method.

On the other hand, the pruning methods that can predetermine the pruning ratio may also have the limitation of empirically adjusted parameters or other limitations. For example, the ratio of channels to prune in each layer may be user-specified such as in SFP (He et al., 2018) and Hrank (Lin et al., 2020), and the compression ratio in each layer is usually selected by empirical studies in these methods. Another type of ratio-predetermined-method is to train and prune the model iteratively until the predefined pruning ratio is achieved, e.g. ThiNet (Luo et al., 2017), which is more complicated than the one-shot pruning.

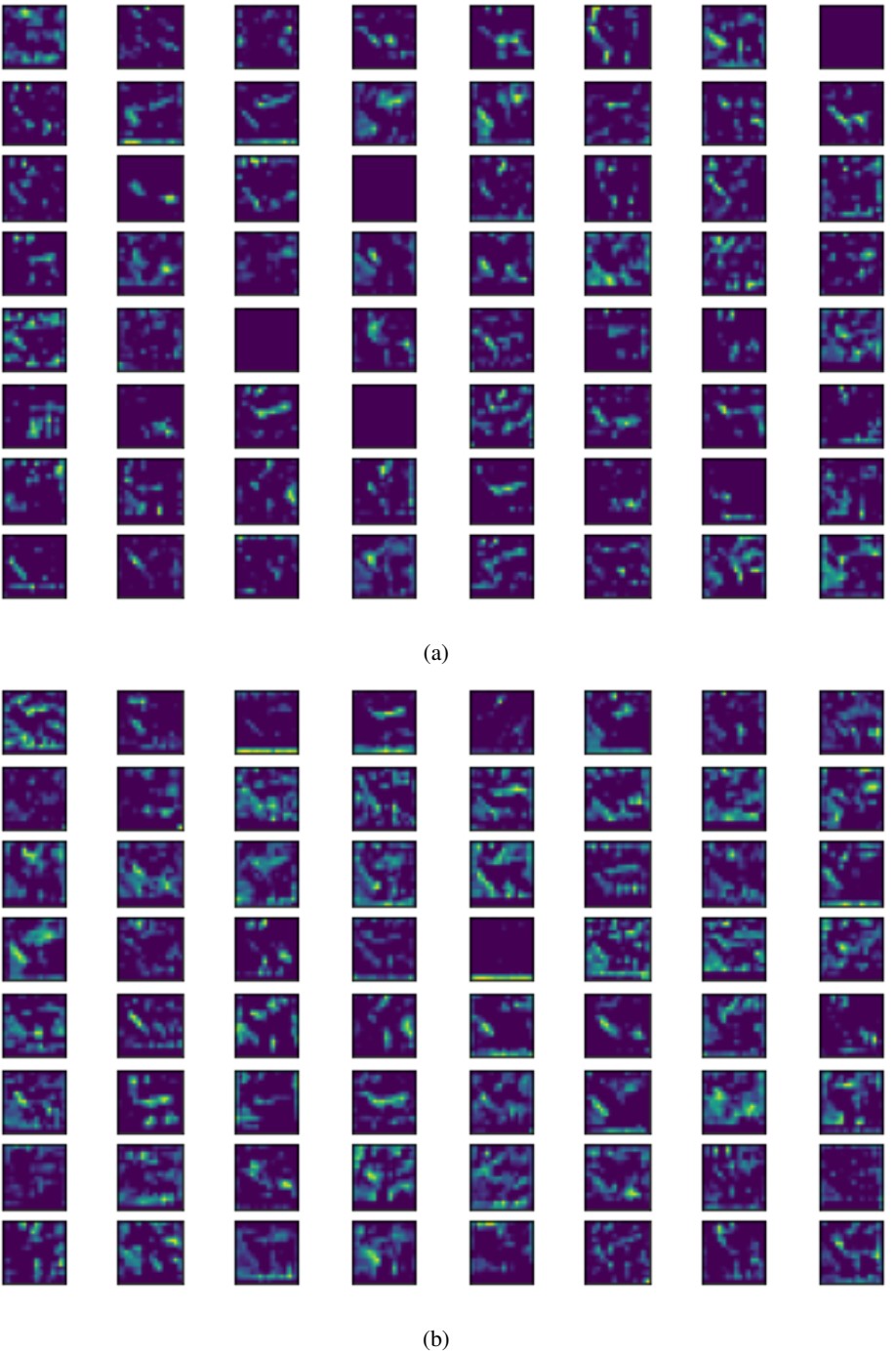

(a)

(b)

Figure 11: Feature maps from the 3rd layer in VGG16 trained with FFR (a) and without FFR (b) on CIFAR-10.

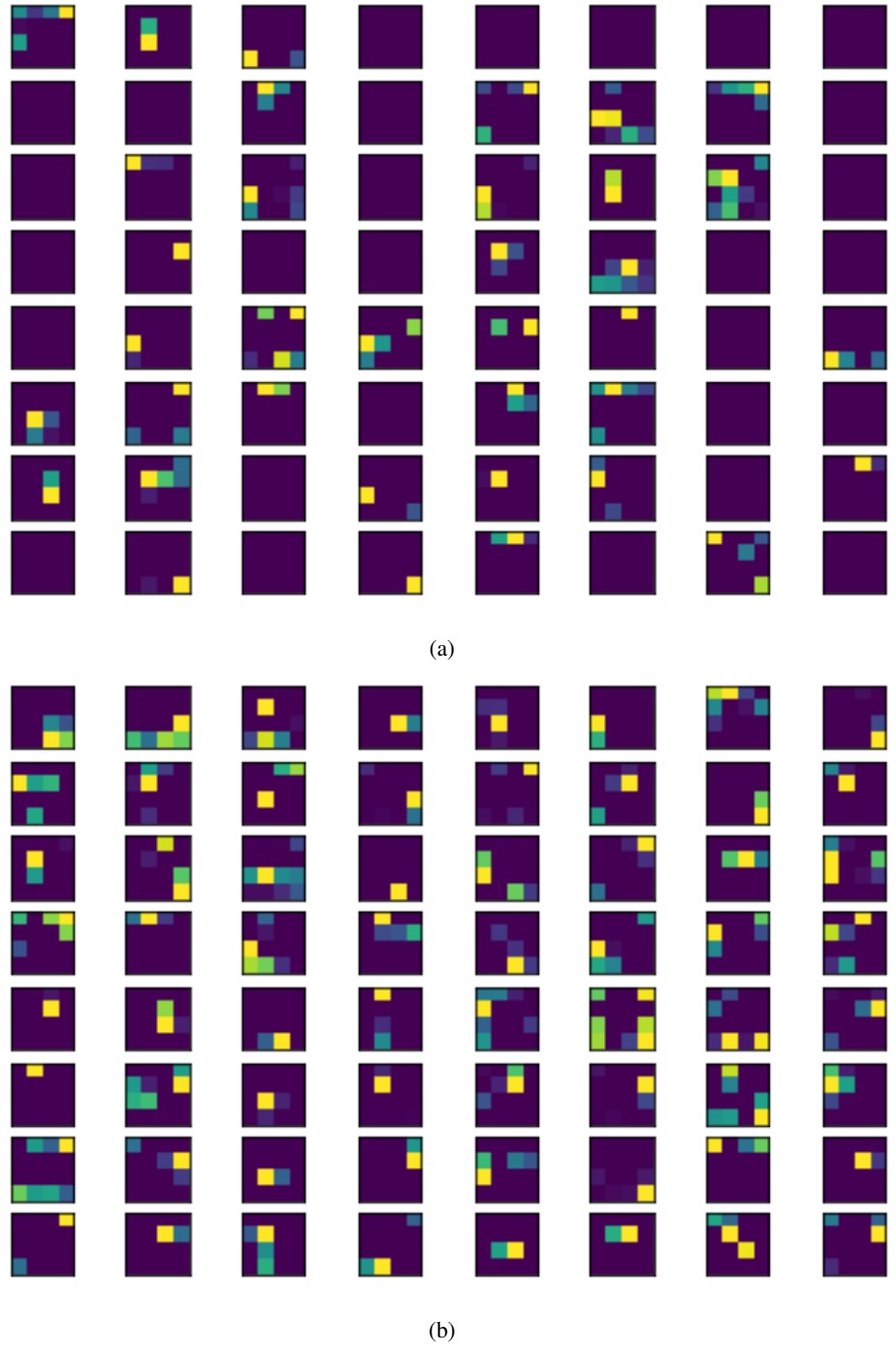

(a)

(b)

Figure 12: Feature maps from the 8th layer in VGG16 trained with FFR (a) and without FFR (b) on CIFAR-10.

