# OpenReview forum: "Feature Flow Regularization: Improving Structured  Sparsity in Deep Neural Networks"
_ICLR.cc/2022/Conference — ICLR 2022 Submitted_

### Official Review · Reviewer_D39H · 2021-10-30

**Correctness:** 3
**Technical Novelty And Significance:** 4
**Empirical Novelty And Significance:** 3
**Recommendation:** 5
**Confidence:** 5

**Main Review:**

Strengths:
1. The feature flow regularization method is inspiring.  The feature flow regularization method (FFR) is simple and effective, and create smooth  consecutive features.
2. Authors provides detailed analysis in Sec.4 to show the method is valid.

Weaknesses:
1. The curvature of two-order regularization needs more illustration. Why we need this item to create sparisty? Besides, ablation studies on the role of length and curvature are absent.
2. As we all know, Convolutional networks were inspired by biological processes. Individual cortical neurons respond to stimuli only in a restricted region of the visual field known as the receptive field. At the bottom level of CNN, the features are basically similar,  such as the edges. The higher, the more features of objects can be extracted (wheels, eyes, etc.), the different advanced features are finally combined into corresponding features, allowing CNNs to accurately distinguish between different objects.
However, with constraints on contingous features and two-order features, the representation power of intermediate features will be reduced significantly. The feature maps of CNNs are used to explore the high-level distinctiveness of inputs. So, more explantations about the FFR will make this paper clear.
3. The results of experiments in Table1,2 have no advantage in contrast to other methods. Expecially in Table.2, the baseline of resnet-50 on Imagenet is 71.56\%, which is significantly lower than 76.1\% officially.  "SCOP: Scientific Control for Reliable Neural Network Pruning" in Neurips 2020 had competitive results, and constastiation should be provided.
4. Table.1 and Table.2 is not consistency. Table.1 reports the error rate, but Table.2 presents the accuracy.

**Summary Of The Paper:**

This paper proposes a regularization strategy for learning sparse deep CNN models. The Feature Flow Regularization (FFR), penalizes first and second order changes in the intermediate features between consecutive layers of the network. The intention of this regularization is to  smooth the evolution of features and make sparse weights.

**Summary Of The Review:**

The paper proposes a regularization loss that penalizes L1 distances between consecutive feature maps that are projected to the same dimension. This loss leads to smooth features and this characteristic is unclear to the target of classification.

---

> ### Author Response · Authors · 2021-11-17
> **Response Part 1**
>
> Thanks for all the comments. We are happy to see that the reviewer agrees with us on the smooth features of our FFR method. We have made further clarifications on the implications of this smoothing characteristic on network pruning as presented below.
>
> ### 1. The curvature of two-order regularization needs more illustration. Why we need this item to create sparsity? Besides, ablation studies on the role of length and curvature are absent.
>
> ### Reply to 1:
> In fact, in addition to the effect that the curvature term keeps the trajectory from bending too much as shown in the main text Sec.3.1 and Fig.1,  the curvature term also encourages more uniform partition of the trajectory from the input to the output, which makes it easier and more accurate to learn with sparse parameters. This effect is associated with the "discretized" trajectory. This can be seen from the following illustrative example: Consider two input-feature-output trajectories
> $$(a): |--|--|, \hspace{6ex} (b): |---|-|$$
> where $'|'$ represents the input, hidden features or the output, $'-'$ denotes one unit $L_1$ distance.
> Here trajectories (a) and (b) have the same length, so the length term cannot distinguish them. Whereas trajectory (a) has smaller "curvature term" than that of (b), thus it is preferred by the curvature term.
> In general, the longer $L_1$ distance between features of adjacent layers, the less possible it is for that layer to learn with sparse parameters. This explanation will be added in the paper.
>
> We have further performed ablation study on hyperparameters $k_1, k_2$ with unequal values as shown in the table below, in addition to the results reported in Appendix in the submission.
> Overall, the curvature term has a stronger impact on the sparsity improvement, and in the mean time, higher error increase, than the length term.
> Compared with applying the length term or the curvature term alone, combining the length term and the curvature term can achieve higher accuracy and sparsity at the same time.
>
> In Table 1, we use the same threshold to prune the network and fine tune 30 epochs and list error, parameters reduction and FLOPs reduction.
>
> We have also plotted accuracy-sparsity trade off curves similar to Fig.4 in the main text for clearer comparison. The new ablation study results and the new figures will be added in the appendix.
>
>
>
> Table 1: Alation study on k1; k2. Pruning results of VGG16 on CIFAR10. ’M’ represents 1e6.
>
> | Hyperparameters | Error   | Param. Reduction | FLOPs Reduction |
> | --- | --- | --- | --- |
> |k_1 =1e-7, k_2 =0 | 6.21  |  75.4  |  34.4  |
> |k_1 =2e-7, k_2 =0 | 6.18  |  77.0  |  37.2  |
> |k_1 =5e-7, k_2 =0 | 6.41  |  84.0  | 44.3   |
> |k_1 =1e-6, k_2 =0 | 7.08  |  91.8  | 62.6   |
> |k_1 =2e-6, k_2 =0 | 8.59  |  95.7  |  70.6  |
> |k_1 =0, k_2 =1e-7 | 6.21  |  77.1  |  37.9  |
> |k_1 =0, k_2 =2e-7 | 6.06  |  79.2  |  38.9  |
> |k_1 =0, k_2 =5e-7 | 6.98  |  92.4  |  67.2  |
> |k_1 =0, k_2 =1e-6 | 8.01  |  96.3  |  80.0  |
> |k_1 =0, k_2 =2e-6 | 9.31  |  98.1  |  90.0  |
> |k_1=k_2=0         | 34.6  | 74.1  | 35.0    |
> |k_1=k_2=1e-7      | 6.45  | 77.5  |  38.6  |
> |k_1=k_2=2e-7      | 6.32  | 85.0  |  47.5  |
>
>
> ### 2. As we all know, Convolutional networks were inspired by biological processes. ... However, with constraints on contingous features and two-order features, the representation power of intermediate features will be reduced significantly. The feature maps of CNNs are used to explore the high-level distinctiveness of inputs. So, more explantations about the FFR will make this paper clear.
>
> ### Reply to 2:
> First of all, the idea of our FFR method is consistent with the building block of the popular ResNet (He et al. 2016). The building block of ResNet is
> $y = \mathcal{F}(x, W)+x$, which can be written as  $y-x = \mathcal{F}(x, W)$,  and this building block does not affect the learning process from low level features to high level features in CNNs. Our FFR method essentially imposes regularization on the residual mapping $\mathcal{F}(x, W)$ to control the parameters $W$, instead of controlling parameters themselves in the available methods.
>
> Pruning is a trade-off between neural network accuracy and efficiency. Experimental results reported in section 5 show that
> FFR is indeed able to learn highly sparse structure while maintaining the representation power of extracting high-level features from low-level features.
>
> Moreover, this can also be demonstrated by feature maps visualization.
> We do have visualization figures of the feature maps of hidden layers in VGG16 with and without FFR on CIFAR-10. These figures show that in the baseline, the features of each layer contain various less informative feature maps that can be pruned, and FFR selects the most representative feature maps and lets other feature maps be zeros.
> We will add these feature maps visualizations of hidden layers in the next version.
>
>
> ### We continue the response in Reponse Part 2.

---

> ### Author Response · Authors · 2021-11-17
> **Response Part 2**
>
> ### 3. The results of experiments in Table1,2 have no advantage in contrast to other methods. Expecially in Table.2, the baseline of resnet-50 on Imagenet is 71.56$\%$, which is significantly lower than 76.1$\%$ officially. "SCOP: Scientific Control for Reliable Neural Network Pruning" in Neurips 2020 had competitive results, and constastiation should be provided.
>
> ### Reply to 3:
> In fact, Table 1 and Fig. 5 on the results on CIFAR-10 show that our FFR method has better performance, i.e., lower errors and larger pruning ratios in terms of parameter and FLOPs, than most of the available methods. This can be better seen from Fig. 5 which plots the data shown in Table 1. In the error vs pruning ratio plots in Fig. 5, our FFR curve is below most of the data points of the available method, especially in the regime of large  parameter and FLOP reductions.
>
> Our FFR method has the advantage of simplicity in the training and pruning.
> With only one-shot pruning and a few epochs fine-tuning
> in FFR, our FFR pruned ResNet50 on the large-scale dataset of ImageNet achieves comparable parameters and FLOPs reductions with the recently proposed pruning methods, as shown in Table 2.
>
> Our FFR trains network from scratch without any pre-training. Such one-shot pruning in FFR is different from those available pruning methods based on well pre-trained networks including the reference mentioned by the reviewer. (Difference between our method and that in this reference will be clarified in the next version.) For the purpose of comparison, we trained the ResNet50 on ImageNet with and without FFR from scratch using the same initialization with only 90 epochs.
> In addition, we are also performing experiments of training ResNet50 on  ImageNet for more epochs and we will report the results once available.
>
>
> ### 4. Table.1 and Table.2 is not consistency. Table.1 reports the error rate, but Table.2 presents the accuracy.
>
> ### Reply to 4:
> Thanks for the suggestion. We will make them consistent by using only errors in these tables in later version.

---

> > ### Comment · Reviewer_D39H · 2021-11-22
> > **Response to Q3**
> >
> > The 76.1% is easily to train with 90 epochs.

---

> > > ### Author Response · Authors · 2021-11-22
> > > **Quick examination of FFR on ImageNet**
> > >
> > > Thanks for the reply.
> > >
> > > Currently, to examine the effect of our FFR method quickly, we are trying the FFR method from a pretrained ResNet50 on ImageNet:  we train ResNet50 with FFR 20 epochs, and
> > > then we prune the ResNet50 and fine tune 20 epochs.
> > >
> > > The top-1 and top-5 accuracy of baseline are 76.15% and 92.87% respectively. We obtained a compact model with top-1 accuracy 72.28%, top-5 accuracy  91.06% and 52.3% parameters reduction, 47.2% FLOPs reduction.
> > >
> > > The results show that FFR enables pruning at a large ratio with 3.87% top-1 accuracy drop and 1.81% top-5 accuracy drop after only 20 epochs of fine-tuning, which is already slightly better than the pruning results of Hrank (Lin et al. 2020).
> > >
> > > We can expect better pruning results (less accuracy drop) after fine tuning more epochs.

---

> > > ### Author Response · Authors · 2021-11-26
> > > **Better pruning results on ImageNet**
> > >
> > >
> > > Under the training setting as described in the last response (train ResNet50 with FFR 20 epochs based on a pretrained ResNet50 on ImageNet),
> > > we continue to fine tune the pruned ResNet50 with appropriate learning rate.
> > >
> > > Currently we have fine tuned ResNet50 with 38 epochs and the top-1 and top-5 accuracy of the pruned ResNet50 are 73.75% and 91.72%, respectively. Baseline top-1 accuracy is 76.15% and top-5 accuracy is 92.87%.
> > > We report the current pruning result and make comparisons with other methods in Table 2 below.
> > >
> > > With a large pruning ratio
> > > (52.3% parameters reduction, 47.2% FLOPs reduction)
> > > but less than 40 epochs fine-tuning,
> > > FFR achieves comparable or better performance than the other state-of-the-art pruning methods.
> > > We can still expect better pruning results (less error increase) after fine tuning more epochs.
> > >
> > >
> > >
> > > Table 2: Pruning results of ResNet50 on ImageNet. '-'represents result not reported. 'M' represents $1e^{6}$.
> > > The top-1 error, top-5 error, parameters and FLOPs of baseline are shown in brackets.
> > > -----------------------------------------------------------
> > > | Method   | Top-1 Err. Increase (%) | Top-5 Err.  Increase (%)|  Param.  Reduction|  FLOPs Reduction   |
> > > | --- | --- | --- | --- | --- |
> > > |Baseline | 0 (23.85)  |  0 (7.13)  |   0% (25.56M)  | 0% (4089M) |
> > > | SSS (Huang & Wang, 2018)   | 4.30  | 2.07 | 38.8%  | 43.0%  |
> > > | DCP (Zhuang et al., 2018)  | 1.06  | 0.61 | 51.5%  |  55.6%  |
> > > | SFP (He et al., 2018)      | 14.01 | 8.27 |   -       | 41.8%   |
> > > | FPGM (He et al., 2019)     |  1.32 | 0.55 |   -       | 53.5%   |
> > > | Hrank (Lin et al., 2020)   | 4.17  | 1.86 |  62.1% | 46.0%   |
> > > | PR (Zhuang et al., 2020)   | 0.52  |  -   |  -        |  54.0%  |
> > > | FFR $(k_1,k_2=5e^{-8})$  | 2.40  | 1.15 |  52.3% | 47.2%   |

---

> > > ### Author Response · Authors · 2021-11-29
> > > **Update: pruning results on ImageNet**
> > >
> > > Now we have fine tuned ResNet50 with 60 epochs and the top-1 and top-5 accuracy of the pruned ResNet50 are 74.03% and 91.78%, respectively.
> > >
> > > We update our pruning results in the table below.
> > >
> > > Table 2: Pruning results of ResNet50 on ImageNet. '-'represents result not reported. 'M' represents $1e^{6}$.
> > > The top-1 error, top-5 error, parameters and FLOPs of baseline are shown in brackets.
> > > -----------------------------------------------------------------------------------------------------------
> > > | Method   | Top-1 Err. Increase (%) | Top-5 Err.  Increase (%)|  Param.  Reduction|  FLOPs Reduction   |
> > > | --- | --- | --- | --- | --- |
> > > |Baseline | 0 (23.85)  |  0 (7.13)  |   0% (25.56M)  | 0% (4089M) |
> > > | SSS (Huang & Wang, 2018)   | 4.30  | 2.07 | 38.8%  | 43.0%  |
> > > | DCP (Zhuang et al., 2018)  | 1.06  | 0.61 | 51.5%  |  55.6%  |
> > > | SFP (He et al., 2018)      | 14.01 | 8.27 |   -       | 41.8%  |
> > > | FPGM (He et al., 2019)     |  1.32 | 0.55 |   -       | 53.5%  |
> > > | Hrank (Lin et al., 2020)   | 4.17  | 1.86 |  62.1% | 46.0%   |
> > > | PR (Zhuang et al., 2020)   | 0.52  |  -   |  -        |  54.0%  |
> > > | FFR $(k_1,k_2=5e^{-8})$  | 2.12  | 1.09 |  52.3% | 47.2%  |

---

### Official Review · Reviewer_wyu7 · 2021-11-01

**Correctness:** 3
**Technical Novelty And Significance:** 2
**Empirical Novelty And Significance:** 3
**Recommendation:** 6
**Confidence:** 3

**Main Review:**

I like the basic premise that encouraging simple mappings in networks 1) doesn't seem to hurt and 2) seems to lead to highly prunable models. I'm not super familiar with the latest developments in the crowded pruning literature and will have to rely on other reviewers on that.

I want to underline a connection to some seemingly unrelated topics. The present idea seems to extend an earlier, frequently-reinvented idea that a network's output should change only a little when the input changes a little. This was probably first coined by Drucker and LeCun as "double backpropagation" [1], and has been heavily used in adversarial learning (input regularization [2]) and GAN training (R1 regularization [3]) in recent years. This was furthermore explicitly relied on in StyleGAN2 [4] for trying to guarantee a simple, well-behaved generator mapping between a latent space and output images (path length regularization). Now, I think your method goes further than any of these by regularizing entire paths through the network instead of just input --> output, thus potentially offering further upside in many applications. That seems like an exciting prospect.

That said, I think the authors need to contrast the method with "trajectory regularization" [5], which was put forward in yet another context, but would seem to do a closely related thing by encouraging simpler representations by regularizing transitions inside a network. It will be important to draw the connection to assess how far beyond pruning the novelty extends.

In my opinion Fig 1 should come sooner than page 4. It is useful mainly as a high-level intuition builder. Figures 3 & 4 are good. Typo: "Strucuted sparsity" (Fig 4).

I did not check all the equations.


[1] Improving generalization performance using double backpropagation, IEEE Trans. Neural Networks, 3(6):991–997, 1992.

[2] Improving the adversarial robustness and interpretability of deep neural networks by regularizing their input gradients, https://arxiv.org/abs/1711.09404

[3] Which training methods for GANs do actually converge?, https://arxiv.org/abs/1801.04406

[4] Analyzing and Improving the Image Quality of StyleGAN, https://arxiv.org/abs/1912.04958

[5] On the Expressive Power of Deep Neural Networks, https://arxiv.org/abs/1606.05336

**Summary Of The Paper:**

This paper deals with network pruning from a fresh perspective. It essentially argues that networks that implement simple mappings are more amenable to pruning. The authors support this claim with experimental evidence. Personally I think this idea is intellectually appealing, and may have uses beyond the network pruning setting. Some concerns related to prior art linger.

**Summary Of The Review:**

An intellectually stimulating take on network pruning, with possible applications elsewhere.

---

> ### Author Response · Authors · 2021-11-19
> **Author response to Reviewer wyu7**
>
> We thank the reviewer for the positive comments with a new prospect of our method and suggestions, and we are glad that you found our idea intellectually appealing and novel. Below we answer your questions:
>
> ### I want to underline a connection to some seemingly unrelated topics. The present idea seems to extend an earlier, frequently-reinvented idea that a network's output should change only a little when the input changes a little. This was probably first coined by Drucker and LeCun as "double backpropagation" [1], and has been heavily used in adversarial learning (input regularization [2]) and GAN training (R1 regularization [3]) in recent years. This was furthermore explicitly relied on in StyleGAN2 [4] for trying to guarantee a simple, well-behaved generator mapping between a latent space and output images (path length regularization). Now, I think your method goes further than any of these by regularizing entire paths through the network instead of just input --> output, thus potentially offering further upside in many applications. That seems like an exciting prospect.
>
> ### That said, I think the authors need to contrast the method with "trajectory regularization" [5], which was put forward in yet another context, but would seem to do a closely related thing by encouraging simpler representations by regularizing transitions inside a network. It will be important to draw the connection to assess how far beyond pruning the novelty extends.
>
>
> First of all, we would like to clarify that our "trajectory", i.e., the feature flow that we define here, is completely different from the "trajectory" defined in the  "trajectory regularization" in reference [5] (in reviewer comments). The trajectory in our method is a 'curve' connecting input, hidden features and output along the neural network. For example, for a feature flow (the trajectory) with "nodes"
>  $ \left\{x_l \right\}_{l=0, 1, 2,\cdots, L} $,
> the "node" $x_0$ is an input data, $x_l$  is a feature of the $l$-th hidden layer for $1\leq l \leq L-1$, and $x_L$ is an output data. Our trajectory reflects the way how data evolves along the neural network.
>
> On the other hand,   [5] (in review comments) defined a trajectory (curve) in the input space and studied the change of this trajectory (mainly its length) as it goes through the hidden layers and eventually reaches the output layer. That is, there is a trajectory at each layer and the trajectory is entirely within this layer in their definition, in contrast to our trajectory that connects different layers. They showed that the length of this trajectory increases exponentially with depth, which reflects the complexity, and this  quantitative measurement  went beyond the earlier studies on network generalization directly from input to output [1-4] (in review comments).
>
> The different definitions of our "trajectory" and that in [5] (in review comments) are based on different purposes and have different implications,  although both methods try to generate smooth trajectories, i.e. for "regularization". The purpose of our FFR is to improve structured sparsity for pruning, whereas in [5] (in review comments), the purpose of their "trajectory" is to improve the generalization ability of the network, i.e.,  a network's output should change only a little when the input changes a little.
>
> Our FFR method is indeed  able to provide novel prospects for other applications beyond pruning, e.g., network generalization and adversarial robustness, as pointed out by the reviewer. In fact, generalization ability of the network can be further explored based on the framework of FFR in which  perturbations at each node on the path connecting different layers are considered, and this will enable the incorporation of information of the entire path that connects  input, hidden features and output along the neural network and the connections between neighboring layers over the path. Theses discussions will be included in the next version of the paper.
>
>
>
>
>
> ### In my opinion Fig 1 should come sooner than page 4. It is useful mainly as a high-level intuition builder. Figures 3, 4 are good. Typo: "Strucuted sparsity" (Fig 4).
>
> ### Reply:
> We will move Fig. 1 to an earlier place in the next version of the paper. Thanks for this helpful suggestion. The typo will be corrected.

---

### Official Review · Reviewer_Ac8a · 2021-11-02

**Correctness:** 3
**Technical Novelty And Significance:** 3
**Empirical Novelty And Significance:** 2
**Recommendation:** 6
**Confidence:** 4

**Main Review:**

The method seems to be original for structured sparsity and pruning and is rather simple. The experimental comparison is with SOTA methods, and the results show that its performance is reasonable.

The paper provides some analysis for the connection of the proposed regularization and sparsity but unfortunately the analysis is not really rigorous. Moreover, there is no discussion and analysis for the curvature term. Even in the experiments, there is no ablation study to dissect the effect of the two regularization terms. Although the presentation of the method and Eqs. 5 and 7 include separate hyperparameters for the two terms, there is no experiment in the paper to investigate unequal values for these hyperparameters.

The experiments are a bit limited, as they are only on two datasets and two architectures. While the results on CIFAR-10 are somewhat encouraging, overall, the results do not seem strongly in favor of the proposed method, especially on ImageNet.

 Also, one disadvantage of the method is that a user cannot select a desired pruning or sparsity ratio, and the sparsity is indirectly controlled by hyperparameters which are not adequately investigated in the paper.


**Summary Of The Paper:**

The paper proposes a new method, called feature flow regularization, for structured sparsity and pruning in deep neural networks. In the proposed method, the length and curvature of trajectories connecting the features of adjacent hidden layers are penalized which is claimed to implicitly result in structured sparsity. Experiments compare the method with several state-of-the-art methods on two datasets and architectures.

**Summary Of The Review:**

The proposed method seems original, but in my view the theoretical analysis and experimental support are not strong enough and need improvement.

---

> ### Author Response · Authors · 2021-11-21
> **Author response to Reviewer Ac8a, Part 1**
>
> Thanks for all the comments. The FFR method proposed in this paper is a novel regularization strategy to improve the structured sparsity and structured pruning from a new perspective of evolution of features along the network, which is simple and effective.   Here we make further clarifications on the method and the analysis and experimental results in the paper. Revised version of the paper will be uploaded soon.
>
> ### 1. The paper provides some analysis for the connection of the proposed regularization and sparsity but unfortunately the analysis is not really rigorous.
>
> In section 4 of this paper, we show that the proposed FFR method encourages structured of features (i.e., for $x_l$ in Eqs. (13)-(16)) in addition to penalizing the structured sparsity of the parameters (i.e., for $W$ in Eqs. (13)-(16)).
>
> We have  provided analysis to show the  essential effects of FFR on the sparsity of hidden features and convolutional parameters. Here we give more details of how FFR improves the structured sparsity, in addition to Eqs. (13)-(16).
>
> For $W \in R^{n \times c \times \omega \times h}, X \in R^{c \times a \times b}, $ assume that
> $W \otimes X \in R^{n \times p \times q}, $ where $ \otimes $ denotes the convolutional operation, and $n,p,q$ are the dimensions of the output.
> We have
> $$|| W \otimes X ||= \sum_{n,p,q} | W_{n,:,:,:} \otimes X_{:,a_p,b_q}  |
> = \sum_{n,p,q} | \sum_c W_{n,c,:,:} \otimes X_{c,a_p,b_q}  |, $$
> where $|| \cdot ||$ denotes elementwise $L_1$-norm and $| \cdot |$ denotes the absolute value, and $X_{:,a_p,b_q} \in R^{c \times \omega \times h }$ represents the subset in features $X$ that produces the $(p,q)$-th entry $(W \otimes X)_{:,p,q}$ in each cannel of the output.
>
> We first show that controlling  $ || W \otimes X ||$ encourages  both $W$ and $X$ to be small. Expanding tensor $W$ into matrix and tensor $X$ into vector, we rewrite the convolutional operation as matrix-vector multiplication:
> $$  || W \otimes X || = || \hat{W} \hat{X} ||,$$
> where  $|| \cdot ||$ is the elementwise $L_1$-norm.
> Thus we have
> $$ \nabla_{\hat{X} }  || \hat{W} \hat{X} || = \hat{W}^{T} {\rm sign} (\hat{W} \hat{X}),$$
>  where ${\rm sign(\cdot)}$ is the sign function elementwise applied to the matrix. It can be calculated that
> $$ \hat{X}^{T} (-\nabla_{\hat{X} }  || \hat{W} \hat{X} ||) =- \hat{X}^{T}  \hat{W}^{T} {\rm sign} (\hat{W} \hat{X}) =- || \hat{W} \hat{X} || \leq 0.  $$
>
> Noticing that the direction of $ \hat{X}$ is the increasing direction of the function $ \frac{1}{2}|| \hat{X} || ^2 = \frac{1}{2} || X ||^2$, or equivalently,  the increasing direction of the norm $ || \hat{X} ||= ||  X ||$,
> the above equation implies that minimizing  $ || W \otimes X ||= || \hat{W} \hat{X} ||$ leads to decrease in $ || X ||$.
> Similarly, we can have the conclusion that  minimizing  $ || W \otimes X ||$ leads to decrease in $  || W ||$.
>
> In the meantime, under the regularization effects, in order to maintain the performance of the neural network as much as possible, the feature maps in $  X $ tend to have only a few large norm feature maps (which capture the most representative information) and many zero norm feature maps,  rather than feature maps with norms distributed evenly.
> One extreme example without expressiveness is noise which is uniform distribution.
> Therefore, the tradeoff between performance and sparsity of the network  leads to high channelwise sparsity in features.
>
> As a result, the convolutional parameters $W$ are trained to have high filterwise sparsity, since
> the filter $W_{n,:,:,:}$ that generates the zero feature map will become zero in training.
>
>
>
> That is why controlling  $ ||  W \otimes X ||$ encourages  $W_{n,:,:,:} $ and $X_{c,:,:} $ to be small,
> which correspond to the parameters $W$ filter-wise sparsity and the feature $X$ channel-wise sparsity, respectively.
>
>
>
> Using the claim above,
> the length term in FFR penalizes $|| (W_l-I) \otimes x_l ||$
> for convolutional block and encourages high channel-wise sparsity of $x_l$ and high filter-wise sparsity of $W_l-I,$ which leads to high structured sparsity of $W_l.$
> For the residual block, the length term in FFR penalizes $|| W_{l,2}\otimes W_{l,2}\otimes x_l ||$ or $|| x_l ||$, so it improves the structured sparsity of the convolutional parameters and the features. See Eqs. (13)-(16).
>
>  We have also proved by experiments that FFR indeed improves the structured sparsity of parameters and features as shown in Fig.3 in section 4, where images (a) and (b) show the results of feature sparsity and image (c) shows a result of parameters $W$  filter-wise sparsity by filter visualization. As shown in Fig.3(a), (b) and Fig.6 (in appendix A.2), the norm of feature (which is the summation of the norm of all feature maps) under FFR is smaller than that of feature without FFR, and feature under FFR has only a few large norm feature maps  and many zero norm feature maps.
>
>
> These new discussions will be added in the next version.

---

> ### Author Response · Authors · 2021-11-21
> **Author response to Reviewer Ac8a, Part 2**
>
> ### 2. Moreover, there is no discussion and analysis for the curvature term.
>
> In fact, in addition to the effect that the curvature term keeps the trajectory from bending too much as shown in the main text Sec.3.1 and Fig.1,  the curvature term also encourages more uniform partition of the trajectory from the input to the output, which makes it easier and more accurate to learn with sparse parameters. This effect is associated with the "discretized" trajectory. This can be seen from the following illustrative example: Consider two input-feature-output trajectories
> $$(a): |--|--|, \hspace{6ex} (b): |---|-|$$
> where $'|'$ represents the input, hidden features or the output, $'-'$ denotes one unit $L_1$ distance.
> Here trajectories (a) and (b) have the same length, so the length term cannot distinguish them. Whereas trajectory (a) has smaller "curvature term" than that of (b), thus it is preferred by the curvature term.
> In general, the longer $L_1$ distance between features of adjacent layers, the less possible it is for that layer to learn with sparse parameters. This explanation will be added in the paper.
>
>
> ### 3. Even in the experiments, there is no ablation study to dissect the effect of the two regularization terms.
> Although the presentation of the method and Eqs. 5 and 7 include separate hyperparameters for the two terms, there is no experiment in the paper to investigate unequal values for these hyperparameters.
>
> We have further performed ablation study on hyperparameters $k_1, k_2$ with unequal values as shown in the table below, in addition to the results reported in Appendix in the submission.
> Overall, the curvature term has a stronger impact on the sparsity improvement, and in the mean time, higher error increase, than the length term.
> Compared with applying the length term or the curvature term alone, combining the length term and the curvature term can achieve higher accuracy and sparsity at the same time.
>
> In Table 1 below, we use the same threshold to prune the network and fine tune 30 epochs and list error, parameters reduction and FLOPs reduction.
>
> We have also plotted accuracy-sparsity trade off curves similar to Fig.4 in the main text for clearer comparison. The new ablation study results and the new figures will be added in the appendix.
>
>
>
> Table 1: Alation study on k1; k2. Pruning results of VGG16 on CIFAR10. ’M’ represents 1e6.
>
> | Hyperparameters | Error   | Param. Reduction | FLOPs Reduction |
> | --- | --- | --- | --- |
> |k_1 =1e-7, k_2 =0 | 6.21  |  75.4  |  34.4  |
> |k_1 =2e-7, k_2 =0 | 6.18  |  77.0  |  37.2  |
> |k_1 =5e-7, k_2 =0 | 6.41  |  84.0  | 44.3   |
> |k_1 =1e-6, k_2 =0 | 7.08  |  91.8  | 62.6   |
> |k_1 =2e-6, k_2 =0 | 8.59  |  95.7  |  70.6  |
> |k_1 =0, k_2 =1e-7 | 6.21  |  77.1  |  37.9  |
> |k_1 =0, k_2 =2e-7 | 6.06  |  79.2  |  38.9  |
> |k_1 =0, k_2 =5e-7 | 6.98  |  92.4  |  67.2  |
> |k_1 =0, k_2 =1e-6 | 8.01  |  96.3  |  80.0  |
> |k_1 =0, k_2 =2e-6 | 9.31  |  98.1  |  90.0  |
> |k_1=k_2=0         | 34.6  | 74.1  | 35.0    |
> |k_1=k_2=1e-7      | 6.45  | 77.5  |  38.6  |
> |k_1=k_2=2e-7      | 6.32  | 85.0  |  47.5  |

---

> ### Author Response · Authors · 2021-11-21
> **Author response to Reviewer Ac8a, Part 3**
>
> ### 4. The experiments are a bit limited, as they are only on two datasets and two architectures. While the results on CIFAR-10 are somewhat encouraging, overall, the results do not seem strongly in favor of the proposed method, especially on ImageNet.
>
> In order to compare FFR pruning results with other SOTA pruning methods, we have performed experiments with VGG16, ResNet56 on CIFAR-10 and ResNet50 on ImageNet. VGGNet and ResNet are two representative deep neural networks, and most pruning methods are reported based on CIFAR-10 and ImageNet, e.g. SFP (He et al. 2018), FPGM (He et al. 2019), Hrank (Lin et al. 2020), PR (Zhuang et al 2020).
>
>
> In fact, experimental results on CIFAR-10 shown in Table 1 and Fig. 5 demonstrate that our FFR method has better performance, i.e., lower errors and larger pruning ratios in terms of parameter and FLOPs, than most of the available methods. This can be better seen from Fig. 5 which plots the data shown in Table 1. In the error vs pruning ratio plots in Fig. 5, our FFR curve is below most of the data points of the available method, especially in the regime of large reductions in parameters and FLOPs.
>
> In addition, we do have test results of our FFR method compared with baseline using ResNet18, ResNet34 on CIFAR-10, CIFAR-100 and Tiny ImageNet.
> We plotted the accuracy-sparsity trade-off curves of FFR compared with the baseline similar to those in Fig.4,
> and these figures also show that FFR improves the structured sparsity and maintains the accuracy in these experiments. These results will be included in the next version of the paper in appendix.
>
>
> Moreover,
> our FFR method has the advantage of simplicity in the training and pruning.
> With only one-shot pruning and a few epochs fine-tuning
> in FFR, our FFR pruned ResNet50 on the large-scale dataset of ImageNet achieves comparable reductions in parameters and FLOPs with the SOTA pruning methods, as shown in Table 2.
> Please note that our FFR trains network from scratch without any pre-training.
> We only trained ResNet50 90 epochs and fine tune the pruned network 30 epochs.
> Such one-shot pruning in FFR is different from those available pruning methods based on well pre-trained networks or more than 100 epochs fine-tuning.
> We are also performing experiments of training and fine-tuning ResNet50 on  ImageNet for more epochs and we will report the results once available.
>
>
>
> ### 5. Also, one disadvantage of the method is that a user cannot select a desired pruning or sparsity ratio, and the sparsity is indirectly controlled by hyperparameters which are not adequately investigated in the paper.
>
> Our FFR method performs  one-shot pruning after training with regularization, and different pruning or sparsity ratios can be achieved by adjusting the hyperparameters instead of direct selecting the pruning ratio.
> Such indirect control of pruning ratio is common in many other one-shot regularization-based pruning methods such as SSS (Huang and Wang 2018) and PR (Zhuang et al. 2020).
> In addition, we can also obtain the desired sparsity using different thresholds to prune the parameters in different layers in our method.
>
> On the other hand, the pruning methods that can predetermine the pruning ratio may also have the limitation of empirically adjusted parameters or other disadvantages.
> For example, the ratio of channels to prune in each layer may be user-specified such as in SFP (He et al. 2018) and Hrank (Lin et al. 2020), and the compression ratio in each layer is usually selected by empirical studies in these methods.
> Another type of ratio-predetermined-method is to train and prune the model iteratively until the predefined pruning ratio is achieved, e.g. ThiNet (Luo et al. 2017), which is more complicated than the one-shot pruning.
>
>
> For the selection of values of the hyperparameters $k_1, k_2$,
> we have already provided a rule  in appendix A.4: The choice of hyperparameters depends on
> the architecture of the DNN and the scale of the input: for the DNN with more layers and higher
> resolution input, we use smaller $k_1, k_2$. We will emphasize it in the main text in the next version.
>
> Moreover, we have also empirically studied the effect of the hyperparameters $k_1, k_2$ on the accuracy after pruning, parameters reduction and FLOPs reduction. As in the reply to the 3rd comment given above, we have performed more tests in the ablation study on the effects of the two  hyperparameters, and the influences of each hyperparameter alone and their combined effect have been clarified. These new discussions will be included in the next version.

---

> ### Comment · Reviewer_Ac8a · 2021-11-27
> **Update**
>
> Thank you for your responses. The ablation study and additional discussions are helpful and have improved the paper. Hence, I have increased the score. Although the experimental support can still be made stronger, I think the paper would be of interest to the community.

---

> > ### Author Response · Authors · 2021-11-28
> > **Thanks for the reply**
> >
> > Thanks for the careful reading and positive comments.

---

### Author Response · Authors · 2021-11-22
**Revised paper submitted**

Dear reviewers,

We have submitted the revised version of the paper. Thanks again for all the comments. We are happy to answer further questions and to have more discussions.

---

### Decision · Program_Chairs · 2022-01-20

**Decision:**

Reject

**Comment:**

The submission proposes "feature flow regularization" during training to enforce (approximately) sparse network weights which can then be post-hoc pruned.  The form of the regularizer is reasonably well motivated, and the method seems interesting.  The reviewers were split on this, with two recommendations for "marginally above" and one for "marginally below" the threshold of acceptance.  I therefore read the paper in detail, in addition to reading the reviews, rebuttal, and private reviewer comments.  The appendix on the sparsity-accuracy tradeoff and its relationship to the hyperparameters k1 and k2 is an interesting experiment, and overall the authors were very engaged in the reviewing process.

In an initial reading, the term trajectory is indeed vague, although it is presented as a definition.  This ambiguity is reflected in the reviewer discussion where in response to Reviewer wyu7, the authors indicate that there are two different meanings of the word in different papers that are being confused.  In a mature presentation, these definitions should probably be given mathematically early on in a formal definition box, but this would require significantly tightening up the mathematical notation early on.

The results table does not show that the proposed method Pareto dominates other methods (accuracy, sparsity, and latency), which themselves are necessarily limited due to the very high number of published papers on network pruning.  Furthermore, some of the selected comparisons appear to be optimizing for different metrics rather than network sparsity, e.g. DCP reports better accuracy for VGG-16 after pruning is applied.  This indicates that the proposed method is somehow in the crowd, but does not seem to show a clear consistent improvement over SOTA.

Analysis in Appendix A.3 does not really depend on which kind of norm is used - the same conclusion will be reached that ||X|| decreases, while structured sparsity, e.g. with expected sparsity rates, is dependent on the kind of norm.  As such, it's OK, but not a particularly specific result.

On the whole, this indicates that the paper is interesting, but borderline with room for concrete improvements that go beyond the scope of a simple refinement for a camera ready presentation.